# Significant contribution of small icebergs to the freshwater budget in Greenland fjords

Soroush Rezvanbehbahani [1,2 ✉], Leigh A. Stearns [1,2], Ramtin Keramati[3], Siddharth Shankar[1,2] & C. J. van der Veen[4]

Icebergs represent nearly half of the mass loss from the Greenland Ice Sheet and provide a distributed source of freshwater along fjords which can alter fjord circulation, nutrient levels, and ultimately the Meridional Overturning Circulation. Here we present analyses of high resolution optical satellite imagery using convolutional neural networks to accurately delineate iceberg edges in two East Greenland fjords. We find that a significant portion of icebergs in fjords are comprised of small icebergs that were not detected in previously-available coarser resolution satellite images. We show that the preponderance of small icebergs results in high freshwater delivery, as well as a short life span of icebergs in fjords. We conclude that an inability to identify small icebergs leads to inaccurate frequency-size distribution of icebergs in Greenland fjords, an underestimation of iceberg area (specifically for small icebergs), and an overestimation of iceberg life span.

[1] Department of Geology, University of Kansas, Lawrence, KS, USA. [2] Center for Remote Sensing of Ice Sheets, University of Kansas, Lawrence, KS, USA. [3] Institute of Computational and Mathematical Engineering, Stanford University, Stanford, CA, USA. [4] Department of Geography and Atmospheric Science, University of Kansas, Lawrence, KS, USA. ✉email: soroushr@ku.edu

Iceberg production is recognized as a self-organized critical process with fractal frequency–size distribution[1,2] resulting in large numbers of small icebergs. Over the last decade, several studies have investigated the size and spatial distribution of icebergs in Greenland and Antarctica using both optical[3,4] and radar imagery[5–7]. The goal of those studies has been to identify or track large icebergs (surface area $A \geq 10^5\ m^2$) with long life spans. While identifying large icebergs in open waters may appear straightforward (due to the sharp contrast between the reflectance properties of ice and water), identifying small icebergs (hereafter icebergs with $A \leq 10^3\ m^2$) is particularly challenging in optical imagery for two main reasons: (1) small icebergs often have a lower contrast in open water and appear as 'dark spots' and (2) aggregates of small icebergs often appear as one large iceberg. Given that the majority of the previous work on iceberg segmentation is done using threshold-based algorithms, small icebergs are either left unidentified or are clustered as large icebergs. The latter may not introduce a significant error in terms of total iceberg area in a fjord, but it skews iceberg frequency–size distributions and consequently melt rate estimates. Although high-resolution images (~0.5 m) have been used in iceberg detections[8], small icebergs are often neglected and their importance has not been investigated.

Here we use high-resolution optical satellite imagery from PlanetLabs with a spatial resolution of ~3 m to segment icebergs in Sermilik and Kangerlussuaq fjords in East Greenland. For precise segmentation of various iceberg sizes, we employ a deep learning approach and train a convolutional neural network, UNet[9] (hereafter P-UNet, to designate the use of UNet on Planet imagery). UNet architecture consists of a downsampling and an upsampling branch for object identification and localization within the image (see Section S2.1). Although applicability of optical imagery in polar regions is limited by cloud cover or lack of solar illumination, these images are substantially easier to use for providing ground truth data required for training. We choose nearly 400 sub-images (up to ~2000 × 2000 m) from Greenland fjords with a varying number of icebergs in each sub-image. We manually annotate a total of 10,000 instances of icebergs in those sub-images and perform elastic deformation and random rotation to augment the training data to a total of 1800 sub-images (over 40,000 iceberg instances, see Section S2.2). We label Planet sub-images from different fjords in east and west Greenland to incorporate variations in our training data. Because identifying iceberg edges is crucial for frequency–size distributions, we apply a weighted cross-entropy loss function that penalizes the false detection of edges five times greater than false detections of non-edge regions. We perform hyperparameter tuning to find the best parameters that minimize the loss function and conduct cross-validation to ensure that the model is not overfitting (see Section S2.2).

In order to compare the effect of resolution and segmentation method on iceberg detection, we acquire near-coincident (same day) Sentinel-2 imagery at ~10 m resolution. Iceberg detection using optical imagery is commonly done using a thresholding scheme; through a manual iterative process, a pixel intensity threshold is defined above which icebergs are detected[4,8]. Following Moyer and others (2019), we use a normalized digital number of 0.13 from the Top of the Atmosphere (TOA) reflection on the near-infrared band (B8) of Sentinel-2 imagery as a threshold for differentiating between icebergs and fjord water (hereafter S-TOA, to designate the use of TOA thresholding on Sentinel-2 imagery). Sentinel-2 tiles are acquired on June 15 and July 7, 2019, for Sermilik and Kangerlussuaq fjords, respectively. The near-coincident image acquisition of Planet and Sentinel-2 imagery (~30 min apart for Sermilik Fjord and <2 h apart for Kangerlussuaq Fjord) allows a close comparison between high

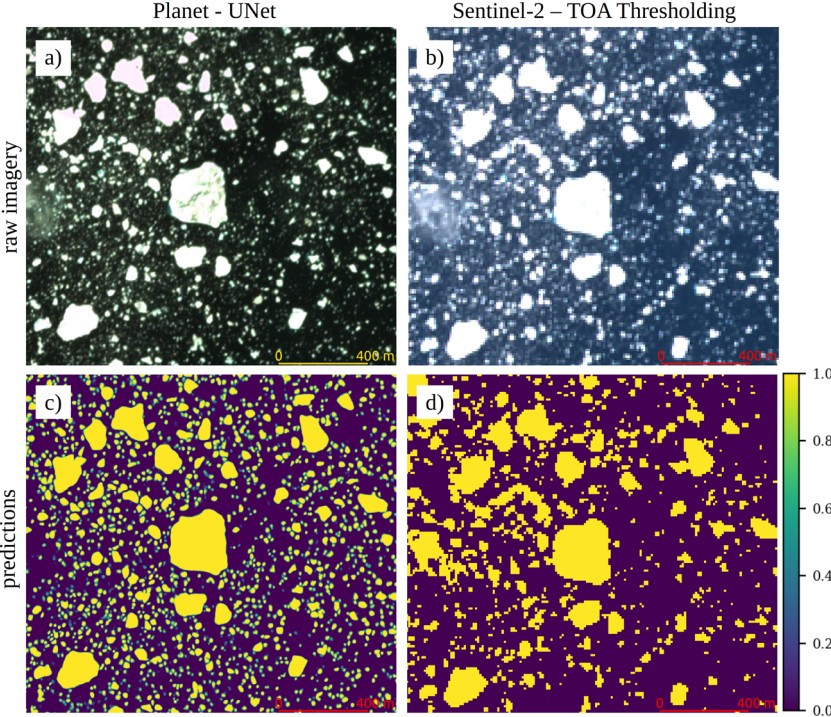

**Fig. 1 Visual comparison of the two segmentation methods.** Examples of UNet performance on Planet imagery **a**, **c** vs. conventional thresholding of top of atmosphere (TOA) of Sentinel-2 imagery **b**, **d**. Both images **a**, **b** are acquired roughly 30 min apart on June 15, 2019. Note the small patch of cloud that is in the left-center of both Planet and Sentinel-2 images, which is falsely detected as an iceberg with thresholding method, while UNet predicts a true negative. The colormap shows the probability of iceberg detections which is set to ≥75% for UNet detections. Panel **a** is from ©Planet Labs Inc. 2020. All Rights Reserved.

(3 m) to moderate resolution (10 m) imagery on iceberg segmentation (Fig. 1). Our analysis is focused on non-mélange regions, because the spectral properties of icebergs within mélange are substantially different from that in open water. All images contain <10% cloud cover.

Note that since our study is focused on exploring the importance of small icebergs, we do not perform UNet on Sentinel-2 imagery. Icebergs that occupy <3–4 pixels in Sentinel-2 imagery ($A < 500$ m$^2$) are extremely hard to identify and manually annotate. This would lead to unreliable training data which hinders robust segmentation of small icebergs. Alternatively, thresholding methods can be applied on high-resolution Planet imagery, however, our analysis using global thresholding and Otsu thresholding methods shows that the frequency–size distribution of icebergs cannot be accurately obtained with generic thresholding methods (see Supplementary Fig. 3), hence, highlighting the superiority of convolutional neural networks for iceberg detection.

## Results

**Iceberg size distribution.** Our results show that the net surface area of detected icebergs can be substantially different between the two approaches. For Sermilik Fjord, segmentation with P-UNet detects $60.4 \pm 7.2$ km$^2$ area of icebergs in the fjord. This surface area is nearly 50% greater than the iceberg surface area obtained with S-TOA ($42.9 \pm 7.7$ km$^2$). Both methods robustly detect large icebergs with $A \geq 10^4$ m$^2$ (~11.5 vs. ~14 km$^2$ for P-UNet and S-TOA, respectively). S-TOA slightly overestimates the total area of large icebergs, due to erroneous merging of clusters of small neighboring icebergs with large ones. However, the total number of small icebergs in Planet imagery is nearly five times greater than those in Sentinel-2 (Fig. 2a, b). This count difference translates to a net surface area of ~4 km$^2$ for P-UNet, as opposed to ~1 km$^2$ for small icebergs using S-TOA. Also, a large portion of differences between P-UNet and S-TOA lies in the range of $10^2 < A < 10^3$ (roughly 2–4 pixels in Sentinel-2 imagery), where icebergs are not sufficiently bright to be detected by thresholding

methods (Figs. 1 and 2). Similar results are obtained for Kangerlussuaq Fjord, however, the area under estimation of S-TOA and abundance of small icebergs is less pronounced. The total number of small icebergs for Kangerlussuaq Fjord is ~13,000 icebergs, more than double the number for S-TOA (Fig. 2c, d).

The frequency–size distribution of icebergs in fjords is often expressed as power-law when iceberg formation is mostly fracture-dominated and log-normal when dominated by iceberg melt[2,4]. However, these designations are without the inclusion of small icebergs. Our analysis confirms that iceberg frequency–size distributions exhibit a power-law, that is $n \propto A^{-\alpha}$, with $n$ the number of icebergs. However, the power-law distribution applies only for icebergs greater than ~300 m$^2$ (Fig. 3). For Sermilik Fjord, S-TOA results in a power-law distribution with $\alpha = 1.29$, as opposed to $\alpha = 1.27$ for P-UNet (Fig. 3a). A smaller difference in power-law exponents is obtained for Kangerlussuaq Fjord (Fig. 3b).

While power-law distribution holds for iceberg sizes greater than ~300 m$^2$, it sharply deviates from power-law at small icebergs, hinting at a critical shift in the dominant processes that control different populations[2,4,10]. For icebergs with $A \leq 300$ m$^2$, the distribution becomes relatively steady. The observed uniformity in frequency of small icebergs is likely due to the fact that small icebergs do not fracture and produce smaller icebergs; instead they shrink and deteriorate only by melting[10]. Identifying the physical meaning of critical iceberg sizes requires further investigation and is beyond the scope of the present work.

The spatial variation in frequency–size distribution is very heterogeneous in both fjords (Fig. 4). Along Sermilik Fjord, both segmentation methods show a notable peak in iceberg sizes ranging ~100–500 m$^2$, with a gradual decrease away from terminus (Fig. 4a, b). However, iceberg size distribution in Kangerlussuaq Fjord exhibits three 'clusters' of iceberg frequency, peaking within the same size range of ~100–500 m$^2$ (Fig. 4c, d). The cause of these cluster formations requires further investigation, however, it is potentially linked to water and wind circulation along the fjord. Both segmentation techniques exhibit similar spatial patterns along the fjord, but the number of iceberg detections are more pronounced in P-UNet. The decrease in

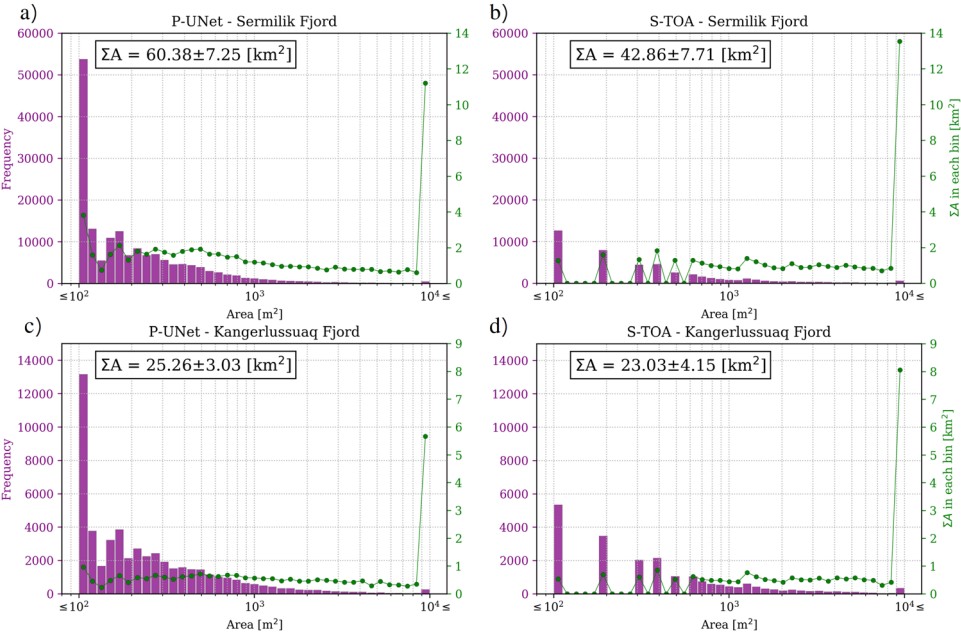

**Fig. 2 Area and frequency of icebergs in two Greenland fjords.** Distribution of icebergs in Sermilik fjord for **a** UNet on Planet imagery, and **b** top of atmosphere thresholding on Sentinel-2 imagery, and similar results for Kangerlussuaq Fjord **c**, **d**. Purple bar charts show the iceberg size distribution (left y-axis) and green lines represent the net area in each bin (right y-axis). Mean weighted error is ~12% for P-UNet and 18% for S-TOA (see Section S2.3).

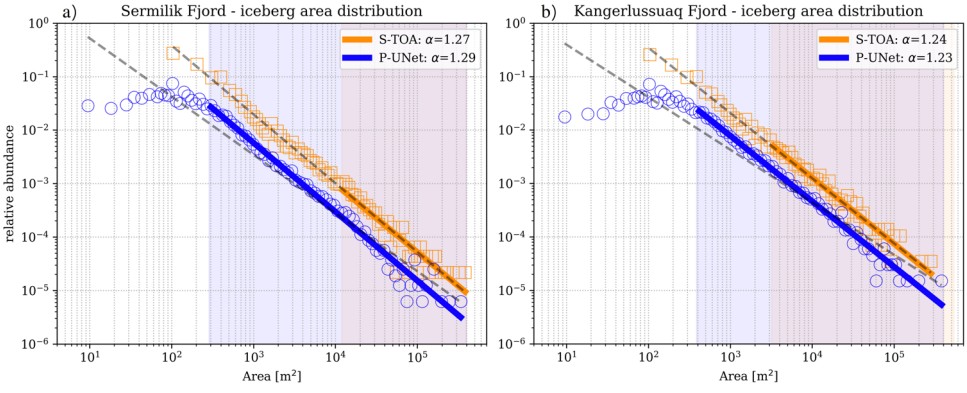

**Fig. 3 Frequency–size distribution of icebergs in the two fjords.** Distribution of iceberg area within **a** Sermilik, and **b** Kangerlussuaq fjords. Solid blue and orange lines correspond to power-law fits (with $\alpha$ exponents) for P-UNet and S-TOA segmentations, respectively. Shaded background areas show the range of data used for colored power-law fits with $A_{min}$ determined by minimizing misfit between data and power-law distribution function[23,24] (see "Methods" section). The black dashed lines represent the power-law fit to the entire data spectrum. The power-law for the entire S-TOA iceberg sizes is very similar to the optimal fit, while the slopes are very different for P-UNet fits. Note that small icebergs ($\leq$300 m$^2$ in P-UNet) clearly deviate from a power-law distribution and are therefore excluded from the fits. Relative abundance is defined as the ratio of number of icebergs in each bin divided by the total number of icebergs.

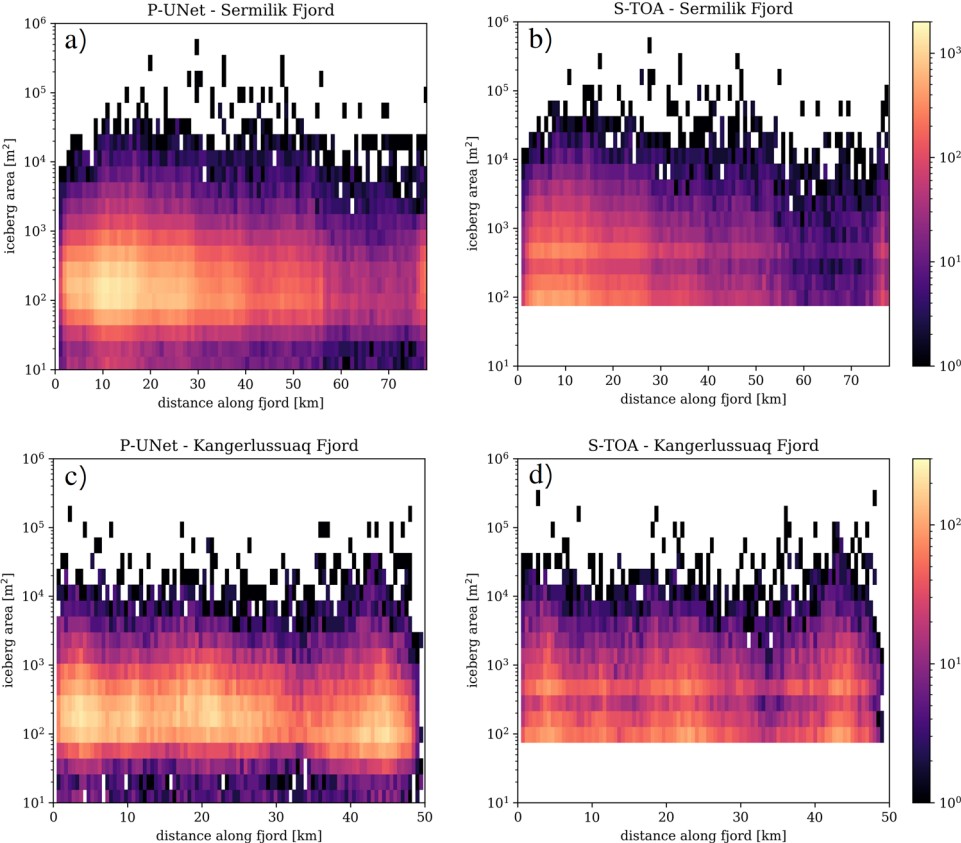

**Fig. 4 Spatial distribution of icebergs in two Greenland fjords.** 2-D histogram of icebergs' size distribution along the fjord for **a, b** Sermilik, and **c, d** Kangerlussuaq fjords for both segmentation methods. Colors show the number of icebergs. *x*-axis starts near the terminus and ends near fjords mouth (see Supplementary Fig. 1). Note that the two fjords have different color ranges.

number of icebergs smaller than ~100 m$^2$ is evident in P-UNet detections for both fjords (Fig. 4a, c).

**Iceberg melt rate and life span.** In order to evaluate the significance of small icebergs in the freshwater budget, we derive a simple set of equations to estimate the freshwater discharge from icebergs. The melt rate is governed by three major processes[11,12]

(i) wave erosion induced by winds, $M_e$, (ii) erosion of the submerged sidewall from buoyancy convection, $M_v$, and (iii) turbulent melt at the bottom of icebergs, $M_b$. Among these processes, wave-induced erosion of submerged sidewalls is at least an order of magnitude greater than the other two[13]. Our derivation incorporates $M_e$ and $M_v$, while $M_b$ is assumed negligible relative to $M_e$. These melt rates can be estimated using fjord water

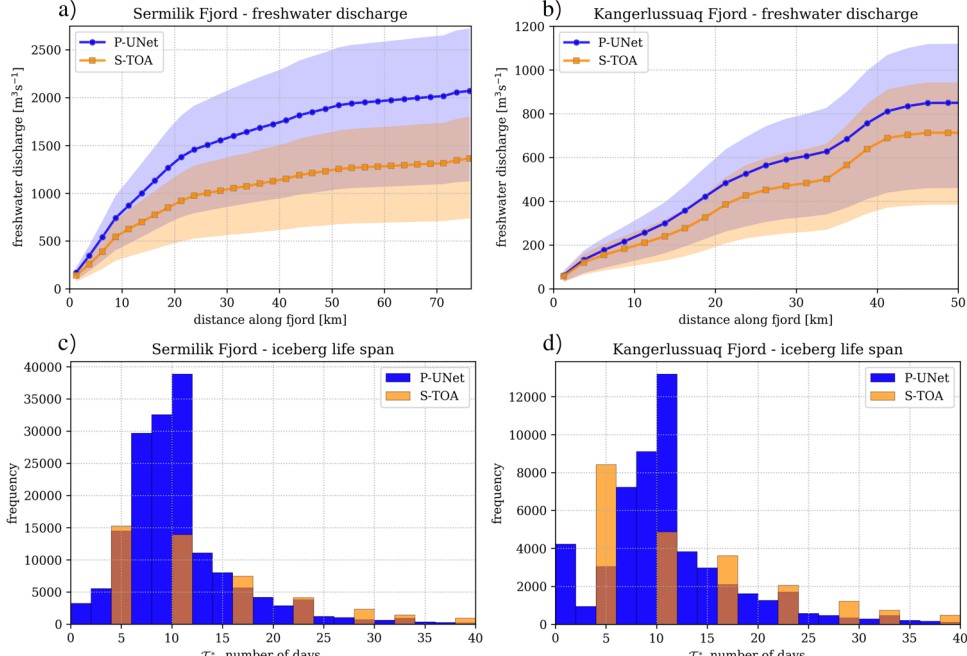

**Fig. 5 Estimates of freshwater discharge from icebergs along fjords and iceberg life span in each fjord.** The distance between melange edge and fjord mouth is divided into 2.5 km segments and melt rate is calculated within each segment. Solid lines in panels **a** and **b** represent melt rate calculations with wind velocity $\overrightarrow{v}_a = 1.7\ \mathrm{m\,s^{-1}}$ which is the mean $\overrightarrow{v}_a$ for June–July 2019, obtained from PROMICE MIT weather station[25]. Colored regions represent one standard deviation of wind velocity $\sigma = 1.2\ \mathrm{m\,s^{-1}}$ (see Section S4). Life span histograms are calculated with $\overrightarrow{v}_a = 1.7\ \mathrm{m\,s^{-1}}$. Note the differences in y-axes.

properties and wind conditions (see Section S4), and are much greater than surface melt rate[13] which is ignored here.

To estimate the width and keel depth of icebergs given their surface area, we use empirical relations that link the longest length of an iceberg ($L$) to iceberg width ($W = L/1.62$) and keel depth ($D = 2.91L^{0.71}$)[13,14]. $L$ is chosen as the maximum length of a rectangular bounding box around an iceberg. This assumption allows the derivation of a time-dependent volume/mass relationship of icebergs (see Section S4) that reads

$$M(t) = \rho_{\mathrm{ice}}1.8[-(M_e + M_v)t + L]^{2.71}, \qquad (1)$$

where ice density is $\rho_{\mathrm{ice}} = 917\ \mathrm{kg\,m^{-3}}$, and $t$ denotes time (in s). The mass of fresh water from iceberg melt is estimated from the difference between $M(t)$ in Eq. (1), and the total iceberg mass $M = \rho_{\mathrm{ice}}LWD = \rho_{\mathrm{ice}}1.8L^{2.71}$. Iceberg life span, $\mathcal{T}^*$, can be readily estimated by setting $M(t) = 0$, therefore, $\mathcal{T}^* = \frac{L_0}{M_e + M_v}$, where $L_0$ is the initial length of an iceberg.

To estimate the spatial distribution of meltwater production, we divide the fjords into 2.5 km segments; within each segment we estimate the melt rate, as well as iceberg life span (Fig. 5a, b). These variables are estimated for both P-UNet and S-TOA segmentations. Our estimates of freshwater mass production using S-TOA match those of ref. [15]. For Sermilik Fjord our discharge estimate along the fjord is ~1310 ± 420 m³ s⁻¹ which is comparable to 1270 ± 735 m³ s⁻¹ for the June–July average in 2017[15], confirming the validity of our freshwater estimates (Fig. 5a). However, the estimated freshwater production using P-UNet in Sermilik Fjord is ~2020 ± 630 m³ s⁻¹ which is more than 50% greater than that of S-TOA. These results demonstrate the significant input of freshwater delivery to Sermilik Fjord due to the large number of small icebergs. The estimated melt rates for Kangerlussuaq Fjord using P-UNet is greater than that of S-TOA. Since the difference between the net iceberg detections are not as great as those of Sermilik Fjord, the freshwater discharge estimates are more similar (Fig. 5b). However, note that S-TOA overestimates the net area of large icebergs ($A > 10^4\ \mathrm{m^2}$, Fig. 2c, d)

which partially explains why the freshwater estimates are closer for Kangerlussuaq fjord.

Our analysis shows that a large portion of icebergs will not live longer than ~10 days. Since the majority of these small icebergs are close to the terminus, we conclude that most icebergs disintegrate before reaching the ocean (similar to findings using Sentinel-2 data for large icebergs[15]). The difference between life span calculations is more notable for Sermilik Fjord (Fig. 5c), and is less pronounced for Kangerlussuaq Fjord (Fig. 5d).

## Discussion
A comparative study of different sources of freshwater input to Sermilik Fjord shows that although the peak of subglacial discharge is greater than iceberg melting, the net freshwater from icebergs exceeds subglacial discharge on an annual scale. The findings of ref. [13] are based on an average June–July melt rate estimates of ~400–600 m³ s⁻¹, nearly 3–5 times smaller than our ~2020 m³ s⁻¹ estimates. Although the analysis of temporal changes of icebergs freshwater distribution is beyond the scope of the present work, if we assume that the contribution of small icebergs is proportionally the same throughout the year, the freshwater budget from iceberg melt can be up to five times greater than some previous estimates[13].

In addition to direct estimation of iceberg freshwater flux from iceberg detections, alternative methods have been developed to infer freshwater discharge from analyzing heat and salt budget in Greenland fjords[16]. Applying these indirect methods to Sermilik Fjord suggests that the net freshwater discharge (iceberg melt and glacier submarine melt) in the summer is ~1200 ± 700 m³ s⁻¹. Although these estimates are for a different time period than our analysis (2011–2013), they are substantially lower than estimates presented in this study. Unlike large icebergs, freshwater delivery from small icebergs is near the surface that may induce a different buoyancy-driven circulation from the freshwater forcing in the fjord. Investigating the oceanographic implications of our findings

requires improved analysis of salt and heat budget of Greenland fjords using the updated estimates of freshwater budget.

We note that in estimating life span and melt rate, we have not considered the stratigraphy of the fjord water and the possibility of having two different water currents at depth. The stratigraphy of fjord water becomes important when the iceberg has a deep keel and portions of the iceberg fall in the boundary between two different water flow regimes in the fjord[17]. In such cases, velocity difference of water currents at depth can substantially alter the melt rate. However, since the focus of our study is the significance of small icebergs, we ignore these fjord properties.

Apart from the glaciologic and oceanographic significance of iceberg detection, they are becoming increasingly important for maritime transport. With the increase of Arctic temperatures and the decline in perennial sea ice coverage, a larger region of the Arctic will be open to marine transportation. Therefore, it is crucial to accurately identify both small and large icebergs in Greenlandic fjords and the North Atlantic ocean, as well as investigating their trajectories[18–20] with high spatial and temporal resolution.

## Methods

**Manual annotation**. We obtain PlanetScope tiles from different fjords in east and west Greenland and divide them into sub-images of up to ~700 × 700 pixels for manual annotation. We label nearly 200 sub-images with a total of about 10,000 individual iceberg annotations. The annotation is done using VGG annotation tool[21]. Data augmentation is done using elastic deformation and random rotation to increase the training data.

**Training and validation**. We use a deep convolutional neural network architecture, UNet[9], for iceberg detection. A weighted cross-entropy loss function is implemented, with iceberg edges weighted five times greater than the rest of the sub-image. This heavy penalization ensures that the network identifies iceberg edges properly. Cross-validation is then performed by training on ~80% and validating on ~20% of the labeled sub-images. We use Adam optimizer[22] to minimize the weighted cross-entropy loss function. The loss function is minimized by a learning rate of $5 \times 10^{-5}$ in 50 epochs with 100 iterations in each epoch. Hyperparameter tuning is performed to ensure the loss is minimized, and a dropout of 0.75 is used to prevent overfitting.

**Power-law distributions**. Power-law fits to empirical data vary based on the choice of linear or log-space bins[4]. Because the iceberg sizes span several orders of magnitude, linear binning is inappropriate. Therefore, we divide the distance between the minimum and maximum size of icebergs into 100 logarithmic bins.

Empirical power-law fits often require a minimum value (i.e., $x_{\min}$) below which the power-law does not apply. In our case, the minimum iceberg size is determined by finding the smallest iceberg size that minimizes the Kolmogorov–Smirnov distance between data and the power-law fit[23]. For Sermilik Fjord, optimized $A_{\min}$ values for P-UNet and S-TOA are 288 and 12,000 m², respectively. For Kangerlussuaq Fjord these values are 387 and 3200 m² for P-UNet and S-TOA. The $A_{\min}$ is calculated using Python 'powerlaw' library[24].

**Iceberg melt rate**. The three dominant iceberg melting processes are formulated as

$$M_e = a_1 |\overrightarrow{v}_a|^{1/2} + a_2 |\overrightarrow{v}_a|, \qquad (2)$$

$$M_v = b_1 T_w + b_2 T_w{}^2, \qquad (3)$$

$$M_b = c |\overrightarrow{v}_w - \overrightarrow{v}_i|^{4/5} (T_w - T_i) L^{-1/5}. \qquad (4)$$

The numerical investigation of ref. [13] shows that $M_e \gg M_v, M_b$, therefore, our analysis only includes $M_e$ and $M_v$. Surface air velocity, iceberg velocity, and surface water velocity are represented by $\overrightarrow{v}_a$, $\overrightarrow{v}_i$, and $\overrightarrow{v}_w$, respectively, and $T_w$ denotes the sea surface temperature (SST) at 4 °C. Since basal melt is ignored in our analysis, the derived formulation is independent of parameters in $M_b$. $\overrightarrow{v}_a$ is chosen as the mean recorded wind velocity during June–July 2019 from PROMICE MIT weather station[25] (see Supplementary Fig. 1). The coefficients to the above equations are reported in Section S4.

Using empirical relations of ref. [14], we derive a time-dependent freshwater production from iceberg melt, where width, $W$, and keel depth, $D$, are expressed as a function of iceberg length, $L$, as $W = \frac{L}{1.62}$ and $D = 2.91 L^{0.71}$, which leads to an expression for volume as $V = 1.8 L^{2.71}$. Because $\frac{dL}{dt} = -M_e - M_v{}^{26}$, integration results in a time-dependent expression for iceberg length as

$$L(t) = -(M_e + M_v)t + L_0, \qquad (5)$$

with $t$ as time (in s) and $L_0$ the initial iceberg length. Applying the above equation in the volume–length relationship and converting volume to mass leads to an equation that expresses the iceberg mass as a function of time as

$$M(t) = 1.8 \rho_{ice} [-(M_e + M_v)t + L_0]^{2.71}. \qquad (6)$$

When $M(t) \to 0$, one can determine the life span of an iceberg ($\mathcal{T}^*$), therefore,

$$\mathcal{T}^* = \frac{L_0}{M_e + M_v}. \qquad (7)$$

## Data availability

Sentinel-2 imagery used in this study are available through Sentinel Hub EO Browser https://apps.sentinel-hub.com/eo-browser/ and Planet imagery are obtained from www.planet.com. Unique IDs are listed in Supplementary Tables 1 and 2.

## Code availability

The codes developed for training the network and performing the analysis are available at https://bitbucket.org/soroushr/planet-unet/src/master/.

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

## Acknowledgements

S.R., L.A.S., and S.S. were supported by NASA grant NNX16AJ90G, and C.J.v.d.V. acknowledges the NSF grant PLR-1543530. We thank Planet Labs for providing imagery for this project through NASA grant NNX16AJ90G, awarded to L.A.S.

## Author contributions

S.R. and L.A.S. designed the study. S.R. and R.K. developed the code for training. S.R. and S.S. implemented the prediction workflow. S.R., L.A.S., S.S., and C.J.v.d.V. contributed to analyzing the results. S.R. led writing of the manuscript with contribution from all authors.

## Competing interests

The authors declare no competing interests.

## Additional information

**Peer review information** Primary handling editors: Heike Langenberg

