## [Peer Review File · Communications Earth & Environment]

Web links to the author's journal account have been redacted from the decision letters as indicated to maintain confidentiality.

28th Apr 20

Dear Dr Rezvanbehbahani,

Your manuscript titled "Deep learning reveals increased freshwater input from small icebergs to Greenland fjords" has now been seen by two reviewers, and I include their comments at the end of this message. They find your work of interest, but some important points are raised. We are interested in the possibility of publishing your study in *Communications Earth & Environment*, but would like to consider your responses to these concerns and assess a revised manuscript before we make a final decision on publication.

We therefore invite you to revise and resubmit your manuscript, along with a point-by-point response that takes into account the points raised. In addition to addressing the reviewers' concerns, we will need you to present robust evidence that small icebergs contribute substantially more freshwater to the two Greenland's fjords that you investigate than previously thought. We will also need the method to be described at a level of detail that allows your results to be reproduced.

We encourage you to investigate whether your insights result from the higher resolution or from the machine-learning-based iceberg segregation method, as requested by both reviewers. However, this is not necessary for publication in *Communications Earth & Environment*; if you are unable to clarify, please ensure that the title and full text reflect that.

Please highlight all changes in the manuscript text file.

Please use the following link to submit your revised manuscript, point-by-point response to the referees' comments (which should be in a separate document to any cover letter) and the completed checklist:

[link redacted]

We hope to receive your revised paper within three months; please let us know if you aren't able to submit it within this time so that we can discuss how best to proceed. If we don't hear from you, and the revision process takes significantly longer, we will close your file. In this event, we will still be happy to reconsider your paper at a later date, as long as nothing similar has been accepted for publication at *Communications Earth & Environment* or published elsewhere in the meantime.

We understand that due to the current global situation, the time required for revision may be longer than usual. We would appreciate it if you could keep us informed about an estimated timescale for resubmission, to facilitate our planning. Of course, if you are unable to estimate, we are happy to

accommodate necessary extensions nevertheless.

Please do not hesitate to contact me if you have any questions or would like to discuss these revisions further. We look forward to seeing the revised manuscript and thank you for the opportunity to review your work.

Best regards,

Heike Langenberg, PhD

Chief Editor
Communications Earth and Environment

On Twitter: @CommsEarth

EDITORIAL POLICIES AND FORMATTING

Editorial Policy: [Policy requirements](https://www.nature.com/documents/nr-editorial-policy-checklist.zip)

Furthermore, please align your manuscript with our format requirements, which are summarized on the following checklist:

[Communications Earth & Environment formatting checklist](https://www.nature.com/documents/commsenv-checklist.pdf)

In the event that the manuscript is accepted we will be providing further guidance on formatting but please ensure that the manuscript generally complies with our house style at this stage. The main points are as follows:

* ABSTRACT: less than 150 words and accessible. It should include the background and context of the work, the phrase 'Here we' (present/show/suggest) to indicate where the description of your own work starts, and then the methods/data, main results and conclusions of the paper.

*The text must be split into:

- INTRODUCTION (<1000 words): includes the background and rationale for the work. The final paragraph should be a brief summary of the main results and conclusions. The results of the current study should only be discussed in this final paragraph.
- RESULTS: split into subheaded sections; ensure that the subheadings are no longer than 60 characters including spaces.
- DISCUSSION: without subheadings.
- METHODS: split into subheaded sections; ensure that the subheadings are no longer than 60 characters including spaces

* SUPPLEMENTARY INFORMATION should be organised logically, with all items labelled as one of the following item types, and cited in the main article:

- Supplementary Figures, labelled and referred to as i.e. Supplementary Figure 1 throughout both the Supplementary Information and the main text
- Supplementary Tables, labelled as above
- Supplementary Notes, labelled as above
- Supplementary Discussion
- Supplementary Methods
- Supplementary References

Data: All Communications Earth & Environment manuscripts must include a section titled "Data Availability" at the end of the Methods section or main text (if no Methods). More information on this policy, and a list of examples, is available at <http://www.nature.com/authors/policies/data/data-availability-statements-data-citations.pdf>.

In addition, Communications Earth & Environment endorses the principles of the Enabling FAIR data project (<http://www.copdess.org/enabling-fair-data-project/>). We ask authors to make the data that support their conclusions available in permanent, publically accessible data repositories. (Please contact the editor if you are unable to make your data available).

If a community resource is unavailable, data can be submitted to generalist repositories such as <https://figshare.com/> or <http://datadryad.org/> Dryad Digital Repository. Please provide a unique identifier for the data (for example a DOI or a permanent URL) in the data availability statement, if possible. If the repository does not provide identifiers, we encourage authors to supply the search terms that will return the data. For data that have been obtained from publically available sources, please provide a URL and the specific data product name in the data availability statement. Data with a DOI should be further cited in the methods reference section.

<http://www.nature.com/authors/policies/availability.html>.

REVIEWER COMMENTS:

Reviewer #1 (Remarks to the Author):

This manuscript presents observations on increasing freshwater input from small icebergs to Greenland fjords, using a deep learning approach. A significant problem is addressed and, while the manuscript does not add that much new information to enhance our understanding of icebergs and their distributions, the paper contributes to our understanding by adding one more piece of information about the contribution of small icebergs to studies of freshwater balance in fjords. The results presented provide useful insights about the contribution of small icebergs to freshwater balance in fjords (which are often neglected) and also highlight the applicability of deep learning approaches to the detection of icebergs. This is especially important since iceberg observations are fairly limited. In addition, the results presented fit well into our current understanding of iceberg size distributions.

However, although the manuscript is very well put together and clearly constructed, I felt that the robustness of the method and the type of data chosen (optical imagery) could have been further discussed. Sometimes it is not entirely clear why to choose a passive sensor (a fact that goes against the usual observed in the literature) and why to use deep learning (advantages and disadvantages). This stands out when we look at the title of the manuscript "Deep Learning reveals...". Also, direct comparisons and inferences made about datasets with different spatial resolutions leave many questions open.

In this sense, the manuscript could be strengthened by putting more effort in a brief analysis / discussion on the standardization of the datasets and also on the robustness of the method (accuracy, precision, false positive and miss rates), which could be added in supplementary material. Furthermore, the paper is well written and clearly laid out, with clear figures. I recommend this paper for publication subject to revisions. I have included some specific comments in the attached PDF file.

Reviewer #2 (Remarks to the Author):

The authors present a comparison of total iceberg area and iceberg size-frequency distribution for two fjords in east Greenland. The iceberg data was derived using a neural network iceberg segmentation algorithm on Planet Labs imagery and is compared to results obtained using a thresholding-based segmentation approach on Sentinel-2 imagery. The manuscript describes the authors' overall approach and results, illustrating the possibility that to date, current estimates of freshwater contributions in Greenlandic fjords due to iceberg melt may be too low. The authors attribute this underestimation to a lack of quantification of the contributions of relatively small (<300m² in area) icebergs, which are abundant in many fjord systems but missed by many methods of iceberg segmentation.

Although this work suggests the possibility of an important contribution to the field and considers some important aspects of automated iceberg segmentation and iceberg melt contributions to fjord freshwater budgets, it fails to address the underlying influence of iceberg segmentation method versus input image spatial resolution to support the claim of the title. By comparing two different segmentation methods on two distinct sets of imagery with different spatial resolutions, it becomes impossible to tease apart the relative impacts of segmentation method versus spatial resolution on

the outcomes. Thus, although I agree with the authors that utilization of advanced computing (e.g. machine learning) is likely to improve our ability to segment icebergs, it does not logically flow from the information presented that to capture icebergs <100m² a different method is needed for segmentation in addition to high (2-3m) resolution imagery.

Overall, the manuscript is decently written, with a few grammatical and clarity issues throughout. It would benefit from a careful review for extraneous language, missing information, and repetitive statements. In many cases, the space saved from removal of repeated ideas could be used to provide the reader with critical information (e.g. assumed velocities used in calculations) and discussion/context that is currently excluded entirely or only included in the supplementary material.

The only other overarching comment I have concerns the lack of quantification in several contexts:

1. quantifications of adjectives (e.g. "small" vs "large" icebergs, "high" resolution imagery). Use of these terms varies highly across individuals and contexts and should be more explicitly stated early on in the manuscript.

2. quantifications of errors (uncertainties in final estimates are provided, but there is no discussion about the influence of various input components on these uncertainties).

3. quantifications of the performance of the neural network. The evaluation criteria described and depicted in the supplementary material are quite helpful, but there is no metric or mention of the algorithm's performance according to the validation dataset, and minimal discussion of the potential uncertainty stemming from operator construction of the training dataset (e.g. one vs multiple operators; consistency throughout the period of manual training).

Communications Earth & Environment

Research Paper: Deep learning reveals increased freshwater input from small icebergs to Greenland fjords

Manuscript Number: commsenv-20-0103-t

Overview:

This manuscript presents observations on increasing freshwater input from small icebergs to Greenland fjords, using a deep learning approach. A significant problem is addressed and, while the manuscript does not add that much new information to enhance our understanding of icebergs and their distributions, the paper contributes to our understanding by adding one more piece of information about the contribution of small icebergs to studies of freshwater balance in fjords.

The results presented provide useful insights about the contribution of small icebergs to freshwater balance in fjords (which are often neglected) and also highlight the applicability of deep learning approaches to the detection of icebergs. This is especially important since iceberg observations are fairly limited. In addition, the results presented fit well into our current understanding of iceberg distributions.

However, although the manuscript is very well put together and clearly constructed, I felt that the robustness of the method and the type of data chosen (optical imagery) could have been further discussed. Sometimes it is not entirely clear why to choose a passive sensor (a fact that goes against the usual observed in the literature) and why to use deep learning (advantages and disadvantages). This stands out when we look at the title of the manuscript “Deep Learning reveals...”. Also, direct comparisons and inferences made about datasets with different spatial resolutions leave many questions open.

In this sense, the manuscript could be strengthened by putting more effort in a brief analysis / discussion on the standardization of the datasets and also on the

robustness of the method (accuracy, precision, false positive and miss rates), which could be added in supplementary material.

Furthermore, the paper is well written and clearly laid out, with clear figures. I recommend this paper for publication subject to revisions. I have included some specific comments below.

Observations:

Several times the author does not make clear what he calls “scene” and “images”. Sometimes, the use of 11 scenes for the Sermilik fjord and 4 scenes for the Kangerlussuaq fjord is mentioned, while sometimes the use of 200 images is mentioned. In that sense, I assumed that images are sub-samples taken from the entire optical scenes. Therefore, some comments may be the result of this misunderstanding. Which I recommend to clarify in the manuscript and supplementary material.

Minor Comments:

L31. Although the author is referring, as I understood, specifically to the detection of icebergs using images in the visible (optical) range. The construction of the sentence (also in the abstract) does not make this statement clear. In the last decade, including in the last 5 years, relevant works based on images in the microwave range (Radar) were published exposing the distribution of icebergs (Greenland and Antarctica) with sizes starting from 0.1km² and inserted in the coastal region, surrounded by sea ice and ice mélange. In this sense, I recommend strengthening the contextualization of the problem and the bibliographic references.

L36. When we consider the additional complexity provided by the dense polar atmosphere. The use of passive sensors, such as optical sensors, generally limit the acquisition of images under the occurrence of clouds or even at night. In this sense, I'm wondering why was the use of images from this class of sensors

chosen in this work? Spatial resolution? Temporal resolution? Viability? Applicability in Fjords? It is curious because it goes in the opposite direction to that observed in the literature for studies on icebergs, sea ice and other structures present in polar regions. Perhaps a sentence is pertinent, exposing the advantages / disadvantages of that choice.

L50. As I understand it, 10,000 instances of icebergs were manually selected and then artificially enlarged with elastic deformation and random rotation. I wonder, although the author carried out a validation step, how much this artificial addition can influence predictive models (overfit). I mean, since the deep neural network will extract the image features autonomously which, due to the nature of the data, can be strongly correlated with aspects of pixel intensity and texture. These aspects (intensity and texture) being poor affected by the applied transformations (mainly rotation). What do you think about the diversity contained in the training set?

L96. The power law observed for the size distribution of the icebergs contained in both fjords are very interesting results (-1.29, -1.27). The computed result is very close to that observed and modeled for icebergs in open water (-3/2), meaning that a large part of smaller icebergs fracture from larger icebergs (Brittle fragmentation). I agree with the fact that smaller icebergs tend to dilute and not fragment, but don't you think that because they are icebergs contained in fjords, part of this distribution may be associated with fragmentation along the coast? Perhaps it would be interesting to add a little more discussion to this distribution since it is a very pertinent result for numerical modeling with iceberg modules.

Major comments:

L54. Although in the complementary material the method is better exposed, I miss more information about basic machine learning metrics such as accuracy, precision, false positive rate and misses. Absence that makes it difficult to assess the real performance of the method in relation to other approaches present in the

literature for the same purpose. As well as, more information about aspects like under and over segmentation would be really pertinent.

L79. As I understood, two sets of images were compared, Planet and Sentinel-2. The two data sets have different spatial resolution (high and moderate resolution). I wonder if the higher resolution set was resampled to the same resolution as the lower resolution set, in order to allow comparison. Because it is expected that the set with higher resolution makes it possible to observe a greater number of icebergs, as well as facilitating the segmentation of nearby objects or clusters. Which makes the comparison unfair, figure 1 makes this situation (different spatial resolution) evident.

General comments regarding the method:

SM. L79. As shown in the supplementary material, 11 scenes were used for Sermilik Fjord and 4 scenes for Kangerlussuaq Fjord, all scenes selected with less than 10% cloud coverage and non-presence of ice mélange. I wonder about the variability and representativeness contained in such reduced set of scenes, especially when applied to a deep learning approach that demands broad training sets for the reliable adjustment of predictive models. Do you believe that the resulting predictive model is robust enough to be applied in different fjords, without the need to return to the training set and again to a new training stage?

SM. L54. Removing the disadvantages in relation to computational cost and development effort, why is not using a pre-processing step considered an advantage of the method? Thinking about the applicability of the method in different data sets coming from different sensors, the pre-processing step helps in the standardization of images that can be susceptible to different sources of noise and failure. In that sense, don't you think that the approach may again be making the method too specific for the fjords in question? Because as I realized the author compares it with previous studies. Perhaps it would be constructive to add some sentences about possible limitations.

Figures:

Figure 5. Perhaps it would be better to make the y axes the same for a and b, and perhaps use a logarithmic scale for c and d. Especially when observing the amplitude in the frequency of icebergs among the Fjords. Also, it might be pertinent to add a sentence about the difference between the size of the sample sets for both Fjords, 11 scenes were used for Sermilik Fjord and 4 scenes for Kangerlussuaq Fjord.

**Authors' response to the reviewers comments
on the manuscript ~~Deep learning reveals
increased freshwater input from small icebergs
to Greenland fjords~~ “Significant contribution of
small icebergs to freshwater budget in
Greenland fjords”**

Manuscript Number: commsenv-20-0103-t

Soroush Rezvanbehbahani*^{1,2}, Leigh A. Stearns^{1,2},
Ramtin Keramati³, Siddharth Shankar^{1,2}, and C.J. van der Veen⁴

¹Department of Geology, University of Kansas, Lawrence, KS 66045

²Center for Remote Sensing of Ice Sheets, University of Kansas, Lawrence, KS 66045

³Institute of Computational and Mathematical Engineering, Stanford University, Stanford, CA 94305

⁴Department of Geography and Atmospheric Science, University of Kansas, Lawrence, KS 66045

We would like to thank Chief Editor Heike Langenberg and both reviewers for their insightful comments/criticism. We believe that their comments have elevated the clarity of the manuscript. We agree with both reviewers that the title “Deep learning reveals...” is rather misleading, because the presented results are a combined outcome of deep learning *and* high resolution imagery (Planet vs. Sentinel-2). Therefore, we have changed the title to better represent the paper's content. In the following, we respond to both reviewers comments separately. Our responses are in blue and the original comments are in black. In our responses, the lines that are directly added to the main text of our manuscript are in bold blue. An additional pdf file which has the changes tracked after revision is submitted with the present response file.

*soroushr@ku.edu

Reviewer #1

This manuscript presents observations on increasing freshwater input from small icebergs to Greenland fjords, using a deep learning approach. A significant problem is addressed and, while the manuscript does not add that much new information to enhance our understanding of icebergs and their distributions, the paper contributes to our understanding by adding one more piece of information about the contribution of small icebergs to studies of freshwater balance in fjords.

The results presented provide useful insights about the contribution of small icebergs to freshwater balance in fjords (which are often neglected) and also highlight the applicability of deep learning approaches to the detection of icebergs. This is especially important since iceberg observations are fairly limited. In addition, the results presented fit well into our current understanding of iceberg size distributions. However, although the manuscript is very well put together and clearly constructed, I felt that the robustness of the method and the type of data chosen (optical imagery) could have been further discussed. Sometimes it is not entirely clear why to choose a passive sensor (a fact that goes against the usual observed in the literature) and why to use deep learning (advantages and disadvantages). This stands out when we look at the title of the manuscript “Deep Learning reveals...”. Also, direct comparisons and inferences made about datasets with different spatial resolutions leave many questions open.

In this sense, the manuscript could be strengthened by putting more effort in a brief analysis / discussion on the standardization of the datasets and also on the robustness of the method (accuracy, precision, false positive and miss rates), which could be added in supplementary material. Furthermore, the paper is well written and clearly laid out, with clear figures. I recommend this paper for publication subject to revisions. I have included some specific comments in the attached PDF file.

Thank you for your comments and very helpful critique. We agree with the majority of reviewers’ comments and we have elaborated the concerns in our detailed responses below.

Overview:

This manuscript presents observations on increasing freshwater input from small icebergs to Greenland fjords, using a deep learning approach. A significant problem is addressed and, while the manuscript does not add that much new information to enhance our understanding of icebergs and their distributions, the paper contributes to our understanding by adding one more piece of information about the contribution of small icebergs to studies of freshwater balance in fjords.

The results presented provide useful insights about the contribution of small icebergs to freshwater balance in fjords (which are often neglected) and also highlight the applicability of deep learning approaches to the detection of icebergs. This is especially important since iceberg observations are fairly limited. In addition, the results presented fit well into our current understanding of iceberg distributions.

However, although the manuscript is very well put together and clearly constructed, I felt that the robustness of the method and the type of data chosen (optical imagery) could have been further discussed. Sometimes it is not entirely clear why to choose a passive sensor (a fact that goes against the usual observed in the literature) and why to use deep learning (advantages and disadvantages). This stands out when we look at the title of the manuscript “Deep Learning reveals...”. Also, direct comparisons and inferences made about datasets with different spatial resolutions leave many questions open.

In this sense, the manuscript could be strengthened by putting more effort in a brief analysis / discussion on the standardization of the datasets and also on the robustness of the method (accuracy, precision, false positive and miss rates), which could be added in supplementary material.

Furthermore, the paper is well written and clearly laid out, with clear figures. I recommend this paper for publication subject to revisions. I have included some specific comments below.

Observations:

Several times the author does not make clear what he calls “scene” and “images”. Sometimes, the use of 11 scenes for the Sermilik fjord and 4 scenes for the Kangerlussuaq fjord is mentioned, while sometimes the use of 200 images is mentioned. In that sense, I assumed that images are sub-samples taken from the entire optical scenes. Therefore, some comments may be the result of this misunderstanding. Which I recommend to clarify in the manuscript and supplementary material.

We agree that the terminology is confusing. We have tried to stick to the nomenclature defined by the organization that has acquired the data. PlanetLabs refers to every swath captured by their satellite as “scene”. However, in accordance with reviewer’s comments and for the sake of consistency, we reworded every reference to the large “scenes” from both Planet and Sentinel-2 imagery as “tiles”, and the tiles are broken to sub-images. The term “imagery” is loosely used to refer to either Sentinel-2 or Planet constellation, and ‘images’ is just a general term while referring to cases such as ‘image processing’. We have edited the entire manuscript carefully and have tried to be as specific and consistent as possible.

Minor Comments:

- **L31.** Although the author is referring, as I understood, specifically to the detection of icebergs using images in the visible (optical) range. The construction of the sentence (also in the abstract) does not make this statement clear. In the last decade, including in the last 5 years, relevant works based on images in the microwave range (Radar) were published exposing the distribution of icebergs (Greenland and Antarctica) with sizes starting from 0.1 km² and inserted in the coastal region, surrounded by sea ice and ice mélange.

In this sense, I recommend strengthening the contextualization of the problem and the bibliographic references.

We have clarified in the abstract as **“Here we use high resolution optical satellite imagery and train...”**. We agree that there has been a lot of recent work using, for example, SAR data for iceberg detection. We have added relevant references in the first paragraph of the manuscript as **“Over the last decade, several studies have investigated iceberg distribution in Greenland and Antarctica using both optical (Sulak and others, 2017; Scheick and others, 2019) and radar imagery (Wesche and Dierking, 2015; Frost and others, 2016; Mazur and others, 2017).”**

- **L36.** When we consider the additional complexity provided by the dense polar atmosphere. The use of passive sensors, such as optical sensors, generally limit the acquisition of images under the occurrence of clouds or even at night. In this sense, I’m wondering why was the use of images from this class of sensors chosen in this work? Spatial resolution? Temporal resolution? Viability? Applicability in Fjords? It is curious because it goes in the opposite direction to that observed in the literature for studies on icebergs, sea ice and other structures present in polar regions. Perhaps a sentence is pertinent, exposing the advantages / disadvantages of that choice.

Although applicability of optical imagery is limited in polar regions due to cloud cover and/or lack of proper solar illumination, they are substantially easier to use for providing ground truth training data. It is significantly more challenging to confidently identify “small” icebergs from microwave data such as SAR imagery. In addition, optical imagery have been around for a much longer period of time, and are therefore more suitable for studying the temporal changes that go back several decades (topic of future work). These are the two main reasons we stayed with optical imagery. We have added a sentence in the second paragraph to elaborate why we have chosen optical imagery as: **Although applicability of optical imagery in polar regions is limited by cloud cover or lack of solar illumination, they are substantially easier to use for providing ground truth data required for training.**

- **L50.** As I understand it, 10,000 instances of icebergs were manually selected and then artificially enlarged with elastic deformation and random rotation. I wonder, although the author carried out a validation step, how much this artificial addition can influence predictive models (overfit). I mean, since the deep neural network will extract the image features autonomously which, due to the nature of the data, can be strongly correlated with aspects of pixel intensity and texture. These aspects (intensity and texture) being poor affected by the applied transformations (mainly rotation). What do you think about the diversity contained in the training set?

Figure 1: Example of augmentation by elastic deformation. *Left* panel shows the original sub-image and *right* panel is after a relatively extreme elastic deformation.

In general data augmentation is extremely valuable for increasing the training data by allowing the network to generalize. The generalization of the network works against overfitting and therefore, makes the model more powerful. Rotating images for data augmentation is a very basic method and its goal is to alter the view point to prevent overfitting on specific directionality. In our case, this *may* lead to overfitting, because it preserves the essential elements of icebergs (e.g., textures and shapes). Augmentation by elastic deformation, on the other hand, resolves such issues because it distorts the shape and texture of the images to the extent that sometimes it is not obvious that two images are the same before and after deformation (see Figure 1). The majority of augmentations that we have included prior to training are using elastic deformation (9 times more elastic deformation than rotations) and therefore are not simple rotations to the same image. We have clarified this in section S2.2 as: **...the ratio of augmented images using elastic deformation to random rotation is 9 to 1.** In addition, training and validation losses were closely monitored during training, and a dropout of 0.75 (i.e., a quarter of neurons randomly ‘switched-off’ from the network) was applied to ensure a continuous decrease of validation and training losses. Therefore, we disagree that augmentation has negatively influenced network performance.

- **L96.** The power law observed for the size distribution of the icebergs contained in both fjords are very interesting results (-1.29, -1.27). The computed result is very close to that observed and modeled for icebergs in open water (-3/2), meaning that a large part of smaller icebergs fracture from larger icebergs (Brittle fragmentation). I agree with the fact that smaller icebergs tend to dilute and not fragment, but don’t you think that because they are icebergs contained in fjords, part of this distribution may be associated with fragmentation

along the coast? Perhaps it would be interesting to add a little more discussion to this distribution since it is a very pertinent result for numerical modeling with iceberg modules.

It is true that collisions of icebergs with the coasts can play a role in fragmentation. To our knowledge, however, such mechanisms are considered significant mostly for large tabular icebergs of Antarctica (e.g., MacAyeal and others, 2008). Regardless of the source of iceberg fracturing, the reviewer’s argument supports what we have proposed in the manuscript, that the observed distribution suggests that dominant iceberg sizes are controlled by fracturing and not melting. In short, while we agree with reviewer’s comment, we believe that it introduces another level of complexity that is not the purpose of the study, but surely deserves further investigation.

Major Comments:

- **L54.** Although in the complementary material the method is better exposed, I miss more information about basic machine learning metrics such as accuracy, precision, false positive rate and misses. Absence that makes it difficult to assess the real performance of the method in relation to other approaches present in the literature for the same purpose. As well as, more information about aspects like under and over segmentation would be really pertinent.

That is a very valid point. The reason we did not provide such metrics in the previous version of our manuscript is that these metrics are problem-specific. For example, if the entire purpose of segmentation is to find the total area of icebergs, then if the neighboring icebergs are lumped together as one iceberg, the error in the total area might be very small (perhaps a few pixels). But if the objective is to provide frequency size distribution, then the hypothetical detection above provides 50% error in the total ‘number’ of icebergs which is very significant for our purpose. Regardless, we agree with both reviewers that lack of such universal metrics prohibits future comparison of other methods against our model. Therefore we have updated these metrics in the main text and expanded in the supplementary material as follows:

In the majority of images that are used for training, non-iceberg regions form the majority of pixels. Therefore, the classification problem we present is highly imbalanced and F1 score is an appropriate classification metric. F1 score is a combination of precision and recall defined as

$$\text{precision} = \frac{TP}{TP + FP}, \quad \text{recall} = \frac{TP}{TP + FN},$$

where (T, F) refers to (true, false), and (P, N) refers to (positive, negative) detections. F1 score can then be calculated as

$$\text{F1} = \frac{2 \times \text{precision} \times \text{recall}}{\text{precision} + \text{recall}}.$$

The closer the F1 score is to 1, the better the segmentation model is. On pixel level, our model results in $TP = 0.814$, $TN = 0.996$, $FP = 0.0038$, and $FN = 0.186$. Given these values $F1 = 0.896$.

- **L79.** As I understood, two sets of images were compared, Planet and Sentinel-2. The two data sets have different spatial resolution (high and moderate resolution). I wonder if the higher resolution set was resampled to the same resolution as the lower resolution set, in order to allow comparison. Because it is expected that the set with higher resolution makes it possible to observe a greater number of icebergs, as well as facilitating the segmentation of nearby objects or clusters. Which makes the comparison unfair, figure 1 makes this situation (different spatial resolution) evident.

It is true that the comparison could be ‘unfair’, since Planet imagery has a better spatial resolution than Sentinel-2. However, the goal of the present work is not to compare different methods or different imagery. We are mostly exploring the glaciologic/oceanographic implications of including small icebergs for evaluating the freshwater budget of Greenland fjords. Therefore, we compare a more advanced method (UNet) with better imagery (Planet), vs. a commonly applied method (thresholding) on a frequently used data (Sentinel-2) in glaciology literature. We did in fact run experiments by resampling Sentinel-2 data from 10 to 3 m resolution to match Planet imagery. However, such interpolation adds a significant blur to the image that more or less ‘contaminates’ threshold-based detections.

General comments regarding the method:

- **SM. L79.** As shown in the supplementary material, 11 scenes were used for Sermilik Fjord and 4 scenes for Kangerlussuaq Fjord, all scenes selected with less than 10% cloud coverage and non-presence of ice mélange. I wonder about the variability and representativeness contained in such reduced set of scenes, especially when applied to a deep learning approach that demands broad training sets for the reliable adjustment of predictive models. Do you believe that the resulting predictive model is robust enough to be applied in different fjords, without the need to return to the training set and again to a new training stage?

We have selected our training images from a number of different fjords in both east and west Greenland. Therefore, we expect a decent level of variability captured in our training dataset, hence, increasing the possibility of applying the same trained model to different fjords. However, our objective in the present work is not to ‘solve’ the issue of iceberg detection using Planet data. The specific scenes we acquired for both Sermilik and Kangerlussuaq fjords visually looked very similar to the scenes we have used to train the network and visual inspection shows that our trained model has performed very well in detecting small icebergs in these fjords. It is true that numerous parameters can influence the radiometric properties of a satellite image, including the angle of solar illumination, atmospheric

aerosol content, and even the constellation orbit (ascending/descending). However, given the small size of Planet sensors, they have less precise radiometry and therefore, such inconsistencies are more likely to occur for Planet imagery than other satellite images such as Sentinel-2 or Landsat-8. Based on our experience with Planet data, there *could* be various visual issues with their data. We did not encounter such inconsistencies in the Planet scenes we used. In order to ‘globally’ train a network that detects icebergs in all Greenland fjords, a significantly larger training data is needed. However, as mentioned earlier, that is not the objective of the present work.

- **SM. L54.** Removing the disadvantages in relation to computational cost and development effort, why is not using a pre-processing step considered an advantage of the method? Thinking about the applicability of the method in different data sets coming from different sensors, the pre-processing step helps in the standardization of images that can be susceptible to different sources of noise and failure. In that sense, don’t you think that the approach may again be making the method too specific for the fjords in question? Because as I realized the author compares it with previous studies. Perhaps it would be constructive to add some sentences about possible limitations.

That is a very legitimate point, and we agree that pre-processing and/or normalization *can* improve the generalizability of the method. We did not perform preprocessing for three main reasons. First, unlike the preprocessing steps for SAR data, pre-processings are often ad-hoc for optical imagery. Second reason is the computational expense of performing preprocessing on such high resolution imagery. Third, as mentioned earlier, applicability of the trained model to all different optical sensors has not been the objective of our work. Providing such model is a work-in-progress and will lead to a stand-alone network that can predict icebergs in all publicly available optical images. The comparisons we have presented against previous studies are for threshold-based segmentation in Sentinel-2 data and not for the UNet training on Planet data. To our knowledge, applying deep neural networks to detect icebergs in Planet imagery has not been done before. We have added the following sentences in the Supplementary Material, section S2.2, to highlight the potential importance of pre-processing: **However, we note that pre-processing the images can increase the likelihood of applying the trained network on other optical imagery. Providing such a network is beyond the purpose of the present manuscript.**

Figures:

- **Figure 5.** Perhaps it would be better to make the y axes the same for a and b, and perhaps use a logarithmic scale for c and d. Especially when observing the amplitude in the frequency of icebergs among the Fjords. Also, it might be pertinent to add a sentence about the difference between the size of the sample sets for both Fjords, 11 scenes were

used for Sermilik Fjord and 4 scenes for Kangerlussuaq Fjord.

We did test what reviewer has suggested, but because the total freshwater discharge at Kangerlussuaq is significantly smaller than that of Sermilik, equalizing the y -axes does not seem appropriate. Presenting the y -axes in c) and d) in log-scale also does not seem useful, as it removes the small variabilities in the tail of the distribution. Hence, we have kept this figure as is, but highlighted the differences in y -axes in the caption.

The number of scenes used for the analysis is not equivalent to the size of icebergs. Sometimes the orientation of the orbit is precisely along a fjord and covers the entire fjord with only two scenes. Therefore we avoid making such statements.

Reviewer #2

The authors present a comparison of total iceberg area and iceberg size-frequency distribution for two fjords in east Greenland. The iceberg data was derived using a neural network iceberg segmentation algorithm on Planet Labs imagery and is compared to results obtained using a thresholding-based segmentation approach on Sentinel-2 imagery. The manuscript describes the authors' overall approach and results, illustrating the possibility that to date, current estimates of freshwater contributions in Greenlandic fjords due to iceberg melt may be too low. The authors attribute this underestimation to a lack of quantification of the contributions of relatively small ($<300\text{m}^2$ in area) icebergs, which are abundant in many fjord systems but missed by many methods of iceberg segmentation.

Although this work suggests the possibility of an important contribution to the field and considers some important aspects of automated iceberg segmentation and iceberg melt contributions to fjord freshwater budgets, it fails to address the underlying influence of iceberg segmentation method versus input image spatial resolution to support the claim of the title. By comparing two different segmentation methods on two distinct sets of imagery with different spatial resolutions, it becomes impossible to tease apart the relative impacts of segmentation method versus spatial resolution on the outcomes. Thus, although I agree with the authors that utilization of advanced computing (e.g. machine learning) is likely to improve our ability to segment icebergs, it does not logically flow from the information presented that to capture icebergs $<100\text{m}^2$ a different method is needed for segmentation in addition to high (2-3m) resolution imagery.

We agree with the reviewer that the title “Deep learning reveals...” is misleading; it is not solely because of better technique, but also due to higher resolution imagery that we have been able to detect increased number of icebergs. Therefore we have modified the title of our manuscript. Deep learning methods for image segmentation have been around for a long time, and specifically UNet architecture has been used since 2015 for image segmentation. The reason we have applied a global thresholding method to Sentinel-2 imagery is to demonstrate that estimates of freshwater budget commonly found in glaciology/oceanography literature can be substantially lower than the actual values. Our objective was not to show the power of deep learning over generic computer vision methods (as the reviewer has also acknowledged), because it has already been demonstrated several times (e.g., Mohajerani and others, 2019, present comparison of different vision algorithms against deep learning for glacier terminus detection in optical images).

Another reason we have not applied UNet on Sentinel-2 imagery is the limitation imposed by its resolution. In order for UNet to detect an iceberg that occupies a single 10×10 m ($=100\text{ m}^2$) pixel in Sentinel-2 imagery, the manual annotation data (labels) *must* include such single-pixel icebergs. However, (ignoring the difficulty of labeling individual pixels) it is almost impossible to reliably assign a class (iceberg vs. water) to a single pixel. Providing unreliable training data is grounds for producing unreliable segmentations from a deep learning algorithm. Therefore,

since the focus of our work was to reliably segment smaller icebergs, we refrained from training UNet on Sentinel-2 imagery.

Overall, the manuscript is decently written, with a few grammatical and clarity issues throughout. It would benefit from a careful review for extraneous language, missing information, and repetitive statements. In many cases, the space saved from removal of repeated ideas could be used to provide the reader with critical information (e.g. assumed velocities used in calculations) and discussion/context that is currently excluded entirely or only included in the supplementary material.

We have read the manuscript to ensure repetitive statements as well as unnecessary information are removed and additional relevant information are provided. For wind velocity previously we used as a rough estimate $\vec{v}_a=3 \text{ m yr}^{-1}$ for all our calculations. However, in the updated draft we used the mean wind speed of June-July 2019 from PROMICE MIT weather station data (van As and others, 2011). Location of the weather station is added to the maps in Fig. S1, and the range of wind velocity deviation is chosen as one standard deviation of wind velocity measurements.

The only other overarching comment I have concerns the lack of quantification in several contexts:

1. quantifications of adjectives (e.g. “small” vs “large” icebergs, “high” resolution imagery). Use of these terms varies highly across individuals and contexts and should be more explicitly stated early on in the manuscript.

Agreed. We have edited the manuscript to ensure these adjectives are explicitly defined where necessary.

2. quantifications of errors (uncertainties in final estimates are provided, but there is no discussion about the influence of various input components on these uncertainties).

We agree with the reviewer that error analysis in the previous version was a bit narrow and we have significantly improved this section. We have added a table in the supplementary material reporting percentage errors of our trained network for different iceberg sizes, along with a weighted average error that is used to report bounds on total iceberg area in Figure 2.

3. quantifications of the performance of the neural network. The evaluation criteria described and depicted in the supplementary material are quite helpful, but there is no metric or mention of the algorithm’s performance according to the validation dataset, and minimal discussion of the potential uncertainty stemming from operator construction of the training dataset (e.g. one vs multiple operators; consistency throughout the period of manual training).

That is a very valid point, and reviewer #1 also pointed out a similar concern. We repeat our comments to reviewer #1 here, but additionally note that all the manual annotation

data was provided by one operator and therefore, there is no random error in providing training data. The reason we did not provide such metrics in the previous version of our manuscript is that these metrics are problem-specific. For example, if the entire purpose of segmentation is to find the total area of icebergs, then if the neighboring icebergs are lumped together as one iceberg, the error in the total area might be very small (perhaps a few pixels). But if the objective is to provide frequency size distribution, then the detection provides 50% error in the total ‘number’ of icebergs which is very significant for our purpose. Regardless, we completely agree with both reviewers that lack of such universal metrics prohibits future comparison of other methods against our model. Therefore we have updated these metrics in the main text and expanded in the supplementary material as follows:

In the majority of images that are used for training, non-iceberg regions form the majority of pixels. Therefore, the classification problem we present is highly imbalanced and F1 score is an appropriate classification metric. F1 score is a combination of precision and recall defined as

$$\text{precision} = \frac{TP}{TP + FP}, \quad \text{recall} = \frac{TP}{TP + FN},$$

where (T, F) refers to (true, false), and (P, N) refers to (positive, negative) detections. F1 score can then be calculated as

$$\text{F1} = \frac{2 \times \text{precision} \times \text{recall}}{\text{precision} + \text{recall}}.$$

The closer the F1 score is to 1, the better the segmentation model is. Our model results in $TP = 0.814$, $TN = 0.996$, $FP = 0.0038$, and $FN = 0.186$. Given these values $\text{F1}=0.896$.

References

- van As, Dirk, Robert S Fausto, Andreas P Ahlstrøm, Signe B Andersen, Morten L Andersen, Michele Citterio, Karen Edelvang, P Gravesen, Horst Machguth, Faezeh M Nick and others, 2011. Programme for Monitoring of the Greenland Ice Sheet (PROMICE): first temperature and ablation records, *Geological Survey of Denmark and Greenland Bulletin*, **23**, 73–76.
- Frost, Anja, Rudolf Ressel and Susanne Lehner, 2016. Automated iceberg detection using high-resolution X-band SAR images, *Canadian Journal of Remote Sensing*, **42**(4), 354–366.
- MacAyeal, Douglas R, Marianne H Okal, Jonathan E Thom, Kelly M Brunt, Young-Jin Kim and Andrew K Bliss, 2008. Tabular iceberg collisions within the coastal regime, *Journal of Glaciology*, **54**(185), 371–386.

- Mazur, Alexey, Anna K Wåhlin and Adam Krezel, 2017. An object-based SAR image iceberg detection algorithm applied to the Amundsen Sea, *Remote Sensing of Environment*, **189**, 67–83.
- Mohajerani, Yara, Michael Wood, Isabella Velicogna and Eric Rignot, 2019. Detection of glacier calving margins with convolutional neural networks: a case study, *Remote Sensing*, **11**(1), 74.
- Scheick, Jessica, Ellyn M Enderlin and Gordon Hamilton, 2019. Semi-automated open water iceberg detection from Landsat applied to Disko Bay, West Greenland, *Journal of Glaciology*, **65**(251), 468–480.
- Sulak, Daniel, David Sutherland, Ellyn Enderlin, Leigh Stearns and Gordon Hamilton, 2017. Iceberg properties and distributions in three Greenlandic fjords using satellite imagery, *Annals of Glaciology*, **58**(74), 92–106.
- Wesche, Christine and Wolfgang Dierking, 2015. Near-coastal circum-Antarctic iceberg size distributions determined from Synthetic Aperture Radar images, *Remote Sensing of Environment*, **156**, 561–569.

12th Aug 20

Dear Dr Rezvanbehbahani,

Your manuscript titled "Deep learning reveals increased freshwater input from small icebergs to Greenland fjords" has now been seen by our reviewers, whose comments appear below. In light of their advice I am delighted to say that we are happy, in principle, to publish a suitably revised version in Communications Earth & Environment under the open access CC BY license (Creative Commons Attribution v4.0 International License).

We therefore invite you to revise your paper one last time to address the remaining concerns of our reviewers. We encourage you to address reviewer 2's remaining concerns, in particular regarding the justification of your methodology and the iceberg size distribution.

At the same time we ask that you edit your manuscript to comply with our format requirements and to maximise the accessibility and therefore the impact of your work.

EDITORIAL REQUESTS:

Please review our specific editorial comments and requests regarding your manuscript in the attached "CommsEarth Final revisions information checklist". Please outline your response to each request in the right hand column.

I also attach an edited version of your first paragraph; please modify as necessary and incorporate into the paper.

SUBMISSION INFORMATION:

In order to accept your paper, we require the files outlined in the attached "CommsEarth Final submission file checklist.pdf"

OPEN ACCESS:

Communications Earth & Environment is a fully open access journal. Articles are made freely accessible on publication under a [CC BY license](http://creativecommons.org/licenses/by/4.0) (Creative Commons Attribution 4.0 International License). This license allows maximum dissemination and re-use of open access materials and is preferred by many research funding bodies.

For further information about article processing charges, open access funding, and advice and support from Nature Research, please visit <https://www.nature.com/commsenv/about/open-access>

At acceptance, the corresponding author will be required to complete an Open Access Licence to Publish on behalf of all authors, declare that all required third party permissions have been obtained and provide billing information in order to pay the article-processing charge (APC) via credit card or

invoice.

Please note that your paper cannot be sent for typesetting to our production team until we have received these pieces of information; therefore, please ensure that you have this information ready when submitting the final version of your manuscript.

[link redacted]

Best regards,

Heike Langenberg, PhD

Chief Editor
Communications Earth and Environment

On Twitter: @CommsEarth

REVIEWERS' COMMENTS:

Reviewer #1 (Remarks to the Author):

I am satisfied with the improvements implemented by the author. Mainly the restructuring of the title, highlighting the real focus of the work. I believe the paper is in good shape for publication.

Only one issue caught my attention during the author's replies:

L79: ... 'We did in fact run experiments by resampling Sentinel-2 data from 10 to 3m resolution to match Planet imagery. '...

I believe it was a mistake in the sentence, but only to reinforce it. The author must have resampled from 3m to 10m (Planet imagery to match Sentinel-2) and not the opposite. Because that way the author would be increasing the spatial resolution of a satellite product and we know that this is not possible. The value contained in the pixel (intensity) is an average of the area covered by that spatial resolution, which in this case is an area of 10m. Therefore, all objects contained in these 10m were reduced to a single average value.

Reviewer #2 (Remarks to the Author):

General Comments:

The authors have put substantial effort into their revisions, and I commend them for this effort because it has greatly improved the manuscript clarity. In particular, their de-emphasis of the machine learning aspect in the title more accurately represents the goals and conclusions of their study. They have adequately described their reasoning for not performing a UNet analysis on the Sentinel imagery (particularly in their response to reviewer comments – this idea could still be more clearly articulated within the manuscript, e.g. within the first paragraph of the text. As currently written, the intro does not differentiate between study objectives and imagery used/methods applied, muddying this important idea). However, it is unclear why they cannot threshold the higher resolution Planet imagery as a point of comparison to differentiate the influence of delineation method from imagery resolution on results. This detracts from their argument that previous investigations needed a different method of analysis to detect small icebergs versus higher resolution optical imagery. In short, I would like to be convinced that the success of UNet to better capture small icebergs is due to the method, not the higher resolution.

Methodology aside, the shift in emphasis to the importance of small icebergs as a significant component of freshwater flux is an important one as the main contribution of this manuscript. To that end, I think the discussion could dive a little deeper. For instance, not all estimates of iceberg flux come from extrapolation of iceberg melt rates, though some of the ones that have suggest total freshwater fluxes from icebergs on the order expected by oceanographic studies to explain the seawater composition in Greenlandic fjords. If these previous melt rate estimates are correct, then the suggestion that they are actually underestimated by a factor of 3-5 could have a significant influence on fjord circulation that should be addressed. If they are not correct, then I would like to see how the freshwater flux estimates derived in this investigation compare with non-iceberg-melt-rate derivations of freshwater flux from icebergs (i.e. the approach for estimating iceberg contributions to freshwater flux based on water mass properties).

With a proper justification for methodology and a slightly expanded discussion, this manuscript will be ready for publication and marks an important contribution towards our understanding of iceberg size distributions and the role of the smallest icebergs in contributing to freshwater flux. Though it looks somewhat long, the list of specific line comments below primarily target formatting and grammar issues that will require minimal effort to modify.

Specific (“line”) comments:

Introduction:

Line 30: remove comma – not a compound sentence

Line 31: iceberg size distributions or iceberg spatial distributions? I would guess the former, but then reference is made to identification and tracking (the latter) before continuing on with a discussion of small icebergs (the former).

Line 33: move “surface area A” from line 35 to here, since it’s the first occurrence

line 40: not a compound sentence (no longer going to point this out).

line 53: PlanetScope imagery has not previously been mentioned.

Line 53: inconsistent comma usage in numbers >1000 (1,000? Not sure what journal requirement is)

Line 57: inconsistent capitalization

line 66: references out of order

line 72: “apart” here suggests that this is referring to time between multiple Sentinel-2 tiles –

consider moving this idea into the next sentence

lines 75+: this would be a good spot to clarify why thresholding is not performed on the higher-resolution Planet imagery

Iceberg size distribution:

Line 80-92: The authors have done an excellent job adding critical context for the difference in total iceberg area cited. However, they do not explain where the full 18 km² of difference in area comes from. If only ~6 km² can be explained by large and small icebergs, then where does the other 12 km² in difference come from? Is it equally or proportionally distributed across other iceberg sizes? Are there enough small-ish (>100 m²) icebergs not detected in the Sentinel imagery to explain the full difference in total area?

lines 97-98 and Figure 3: inconsistent references to alpha as positive or negative (in text alpha is positive, in figures it is shown as negative).

Lines 101-102: This idea should be cited

Lines 93+: what about lognormal distributions, which are frequently used to quantify iceberg size distributions where melt dominates (versus fracture)?

Icebergs melt rate and life span:

Section heading is grammatically incorrect

How is iceberg length calculated?

What is the influence of things like assumed ice density on the total magnitude of these results?

Code Availability:

I applaud the authors for making their code publicly available via bitbucket, but I question the usability of the code by a new user given the lack of information on system/package requirements or instructions for running it easily visible on the page (e.g. in the readme).

Methods:

Line 204: misspelling (Kolmogorov)

lines 209-210: if $M_e \gg M_v$, then why is M_v included?

line 216: D is presented as both height (within the main text) and draft (in methods)

Figures:

Figure 4: would it be possible to show results as distance from the terminus rather than as normalized bins? This would make it easier to compare the results with Figure 5.

Any explanation for the seeming absence of icebergs around 0.8 in Sermilik Fjord? I'm curious if it aligns with any geologic features or has another explanation.

References:

Great job standardizing your references!
line 299: inconsistent number of authors listed

Supplementary material:

A few typos but overall great additions that significantly improve the manuscript.

line 20: I'm curious why the authors chose not to mention that they focus on open-water areas and use cloud free imagery in the main text.

Figure S2: what is the grid scale? To my eye, it looks like some small bergy-bits are excluded in the manual annotation.

Figure S4: in merging, how are overlapping pixels treated? averaged? max?

Authors' response to the reviewers comments on the manuscript "Significant contribution of small icebergs to **the** freshwater budget in Greenland fjords" – second revision

Manuscript Number: COMMSENV-20-0103B

Soroush Rezvanbehbahani*^{1,2}, Leigh A. Stearns^{1,2},
Ramtin Keramati³, Siddharth Shankar^{1,2}, and C.J. van der Veen⁴

¹Department of Geology, University of Kansas, Lawrence, KS 66045

²Center for Remote Sensing of Ice Sheets, University of Kansas, Lawrence, KS 66045

³Institute of Computational and Mathematical Engineering, Stanford University, Stanford, CA 94305

⁴Department of Geography and Atmospheric Science, University of Kansas, Lawrence, KS 66045

We thank both reviewers for their constructive criticism. In particular, careful comments by reviewer #2 greatly elevated the clarity of this manuscript. In the following, we respond to both reviewers comments separately. Our responses are in blue and the original comments are in black. In our responses, the lines that are directly added to the main text of our manuscript are in bold blue. An additional pdf file which has the changes tracked after revision is submitted with the present response file.

Reviewer #1

I am satisfied with the improvements implemented by the author. Mainly the restructuring of the title, highlighting the real focus of the work. I believe the paper is in good shape for publication.

Only one issue caught my attention during the author's replies: L79: ... 'We did in fact run experiments by resampling Sentinel-2 data from 10 to 3m resolution to match Planet imagery.

' ...

*soroushr@ku.edu

I believe it was a mistake in the sentence, but only to reinforce it. The author must have resampled from 3m to 10m (Planet imagery to match Sentinel-2) and not the opposite. Because that way the author would be increasing the spatial resolution of a satellite product and we know that this is not possible. The value contained in the pixel (intensity) is an average of the area covered by that spatial resolution, which in this case is an area of 10m. Therefore, all objects contained in these 10m were reduced to a single average value.

We agree that a proper way would be to resample from a higher resolution to a coarser resolution, and the fact that resampling from 10 m to 3 m may introduce additional false positives/negatives to the analysis. However, since our focus was to specifically evaluate the importance of small icebergs, we resampled from 10 m to 3 m which, as the reviewer has suggested, is not a proper resampling. This was one of the experiments we conducted and the results are not reported nor discussed in the manuscript.

Reviewer #2

General Comments:

The authors have put substantial effort into their revisions, and I commend them for this effort because it has greatly improved the manuscript clarity. In particular, their de-emphasis of the machine learning aspect in the title more accurately represents the goals and conclusions of their study. They have adequately described their reasoning for not performing a UNet analysis on the Sentinel imagery (particularly in their response to reviewer comments – this idea could still be more clearly articulated within the manuscript, e.g. within the first paragraph of the text. As currently written, the intro does not differentiate between study objectives and imagery used/methods applied, muddying this important idea).

We have elaborated on this differentiation between imagery/methods in the text. We added a new paragraph (4th) to the first part of the manuscript after the abstract that discusses our comparisons that show UNet performs better than common thresholding method on Planet data and therefore, a robust justification why UNet should be used in our study.

However, it is unclear why they cannot threshold the higher resolution Planet imagery as a point of comparison to differentiate the influence of delineation method from imagery resolution on results. This detracts from their argument that previous investigations needed a different method of analysis to detect small icebergs versus higher resolution optical imagery. In short, I would like to be convinced that the success of UNet to better capture small icebergs is due to the method, not the higher resolution.

In order to evaluate the performance of simple thresholding against UNet on Planet data, we performed a simple global thresholding of 0.2 on the blue band of Planet imagery on validation dataset (see the updated Figure S3). In addition to that, we performed thresholding using Otsu filter on the same validation dataset. Unlike global thresholding, Otsu thresholding is adaptive

per image, and the threshold varies by image and is obtained by finding the intensity of local minima in a bi-modal image histogram. There are several other ‘generic’ thresholding methods in computer vision literature, but these two widely-used methods are sufficient to demonstrate the superiority of UNet against generic vision algorithms.

Methodology aside, the shift in emphasis to the importance of small icebergs as a significant component of freshwater flux is an important one as the main contribution of this manuscript. To that end, I think the discussion could dive a little deeper. For instance, not all estimates of iceberg flux come from extrapolation of iceberg melt rates, though some of the ones that have suggest total freshwater fluxes from icebergs on the order expected by oceanographic studies to explain the seawater composition in Greenlandic fjords. If these previous melt rate estimates are correct, then the suggestion that they are actually underestimated by a factor of 3-5 could have a significant influence on fjord circulation that should be addressed. If they are not correct, then I would like to see how the freshwater flux estimates derived in this investigation compare with non-iceberg-melt-rate derivations of freshwater flux from icebergs (i.e. the approach for estimating iceberg contributions to freshwater flux based on water mass properties).

We agree that the oceanographic implications of our findings are substantial. However, we believe that these implications require thorough modeling and analysis and is beyond the scope of our paper. We have added a short paragraph acknowledging the alternative methods for estimating freshwater budget, but we avoid a detailed discussion without conducting a proper oceanographic investigation. The added paragraph reads: **In addition to direct estimation of iceberg freshwater flux from iceberg detections, alternative methods have been developed to infer freshwater discharge from analyzing heat and salt budget in Greenland fjords (Jackson and Straneo, 2016). Applying these indirect methods to Sermilik Fjord suggests that the net freshwater discharge (iceberg melt and glacier submarine melt) in the summer is $\sim 1200 \pm 700 \text{ m}^3\text{s}^{-1}$. Although these estimates are for a different time period than our analysis (2011–2013), they are substantially lower than estimates presented in this study. Unlike large icebergs, freshwater delivery from small icebergs is near the surface that may induce a different buoyancy-driven circulation from the freshwater forcing in the fjord. Investigating the oceanographic implications of our findings requires improved analysis of salt and heat budget of Greenland fjords using the updated estimates of freshwater budget.**

With a proper justification for methodology and a slightly expanded discussion, this manuscript will be ready for publication and marks an important contribution towards our understanding of iceberg size distributions and the role of the smallest icebergs in contributing to freshwater flux. Though it looks somewhat long, the list of specific line comments below primarily target formatting and grammar issues that will require minimal effort to modify.

Specific (“line”) comments:

Introduction:

- Line 30: remove comma – not a compound sentence

Done.

- Line 31: iceberg size distributions or iceberg spatial distributions? I would guess the former, but then reference is made to identification and tracking (the latter) before continuing on with a discussion of small icebergs (the former).

We are referring to both distributions. We have modified as **...have investigated size and spatial distribution of icebergs in Greenland...**

- Line 33: move “surface area A” from line 35 to here, since it’s the first occurrence

Done.

- line 40: not a compound sentence (no longer going to point this out).

Done.

- line 53: PlanetScope imagery has not previously been mentioned.

We have moved the details of PlanetScope to Methods and SI.

- Line 53: inconsistent comma usage in numbers >1000 (1,000? Not sure what journal requirement is)

All are made consistent (e.g. 2000).

- Line 57: inconsistent capitalization

Fixed.

- line 66: references out of order

Fixed.

- line 72: “apart” here suggests that this is referring to time between multiple Sentinel-2 tiles – consider moving this idea into the next sentence

Agreed. We reworded these two sentences to make sure “apart” is referring to time mismatch in image acquisition from different sensors.

- lines 75+: this would be a good spot to clarify why thresholding is not performed on the higher-resolution Planet imagery

Agreed. We have added the following: **Alternatively, thresholding methods can be applied on high resolution Planet imagery, however, our analysis using global**

thresholding and Otsu thresholding methods shows that the frequency-size distribution of icebergs cannot be accurately obtained with generic thresholding methods (see Fig. S3), hence, highlighting the superiority of convolutional neural networks for iceberg detection.

Iceberg size distribution:

- Line 80-92: The authors have done an excellent job adding critical context for the difference in total iceberg area cited. However, they do not explain where the full 18 km² of difference in area comes from. If only ~6 km² can be explained by large and small icebergs, then where does the other 12 km² in difference come from? Is it equally or proportionally distributed across other iceberg sizes? Are there enough small-ish (>100 m²) icebergs not detected in the Sentinel imagery to explain the full difference in total area?

As evident in Figure 2, several bins in the range of 10² – 10³ are empty for S-TOA results, while there are many icebergs detected in those bins based on P-UNet results. This shows that the lack of ‘single-pixel’ detections in S-TOA is not the only reason for such area differences. There are many icebergs that may occupy only a few pixels in S-TOA, but are not bright enough to be captured by thresholding. If thresholding is lowered to the point that those icebergs are detected, then the false positive rate of iceberg detection will be so high that the S-TOA segmentation results would be totally unreliable and total iceberg area egregiously high. This can visually be seen in Figure 1 as well. Based on this suggestion, we modified our definition of “small” iceberg in text to $A < 10^3$ m² and clarified as follows: **Also a large portion of differences between P-UNet and S-TOA lies in the range of $10^2 < A < 10^3$ (roughly 2 to 4 pixels in Sentinel-2 imagery) where icebergs are not sufficiently bright to be detected by thresholding methods (Figs. 1 and 2).**

- lines 97-98 and Figure 3: inconsistent references to alpha as positive or negative (in text alpha is positive, in figures it is shown as negative).

We have made references to α consistent with the equation $n \propto A^{-\alpha}$. Therefore, we made the α labels in Figure 3 positive.

- Lines 101-102: This idea should be cited

Marko (1996), Kirkham and others (2017), and Scheick and others (2019) are added.

- Lines 93+: what about lognormal distributions, which are frequently used to quantify iceberg size distributions where melt dominates (versus fracture)?

We have clarified as follows: **The frequency-size distribution of icebergs in fjords is often expressed as power-law when iceberg formation is mostly fracture-**

dominated and log-normal when dominated by iceberg melt (Kirkham and others, 2017; Scheick and others, 2019).

Icebergs melt rate and life span:

- Section heading is grammatically incorrect

We changed it to “Iceberg melt rate and life span”.

- How is iceberg length calculated?

We have clarified as “*L* is chosen as the maximum length of a rectangular bounding box around an iceberg.”

- What is the influence of things like assumed ice density on the total magnitude of these results?

The range of values that are used for ice density in glaciology literature is quite narrow, perhaps 910–920 kg m⁻³. So we don’t anticipate any notable difference if higher/lower values of ice density is used. Even if a different value for ice density is used, it will be applied to all icebergs for both fjords of P-UNet and S-TOA, and therefore, the relative difference between the two detection methods will not change.

Code Availability:

I applaud the authors for making their code publicly available via bitbucket, but I question the usability of the code by a new user given the lack of information on system/package requirements or instructions for running it easily visible on the page (e.g. in the readme).

We have cleaned up some sections of the code and added comments to make the code more usable. We have also added a requirements.txt file to the repository that shows which packages are needed for the repository. At present all components of the code that we used for this study are available. Our objective is to make this code usable for a variety of earth science disciplines. The present code uses TensorFlow 1.13, however, during the completion of this work TensorFlow 2.x was released and is substantially different from TensorFlow 1.x which is no longer supported. Hence, we are working on migrating the current code structure to TF 2.x and create a more user friendly tool for the community.

Methods:

- Line 204: misspelling (Kolmogorov)

Fixed.

- lines 209-210: if $M_e \gg M_v$, then why is M_v included?

Including or excluding M_v or M_b will not have a substantial impact on the present melt rate/life span estimates. But despite the simplified assumptions we have made to derive the melt rate equations, it is preferred to keep as many melt rate variables as possible.

- line 216: D is presented as both height (within the main text) and draft (in methods)

D refers to iceberg keel depth (or draft). We have made the references to D as keel depth (instead of height) and notations (D instead of H) consistent throughout the text and SI.

Figures:

Figure 4: would it be possible to show results as distance from the terminus rather than as normalized bins? This would make it easier to compare the results with Figure 5.

Great suggestion. Done.

Any explanation for the seeming absence of icebergs around 0.8 in Sermilik Fjord? I'm curious if it aligns with any geologic features or has another explanation.

We did not find any geologic feature or change in fjord bathymetry in that region so we cannot conclude that the observed absence of icebergs is linked to any geologic feature. Moreover, most of the icebergs are small and do not have a deep keel depth to be influenced by subtle changes in the bathymetry. It could possibly be due to changes in fjord water circulation or wind direction that may have caused that moderately iceberg free zone.

References:

Great job standardizing your references! line 299: inconsistent number of authors listed

It is unclear which reference the reviewer is referring to since our manuscript is less than 299 lines. However, we have revised the references and resolved inconsistencies.

Supplementary material:

A few typos but overall great additions that significantly improve the manuscript.

line 20: I'm curious why the authors chose not to mention that they focus on open-water areas and use cloud free imagery in the main text.

We added these information to the third paragraph of the manuscript.

Figure S2: what is the grid scale? To my eye, it looks like some small bergy-bits are excluded in the manual annotation.

Great observation! This figure was from an earlier version of the manuscript where the focus was not on small icebergs. We have updated this figure to a one that has complete annotations that were used in the training. Image size is 768×768 pixels, equivalent to $\sim 2300 \times 2300$ m. We added the image size in the caption of this figure.

Figure S4: in merging, how are overlapping pixels treated? averaged? max?

We have chosen the maximum pixel value. We added this to the updated flowchart in Figure S4.

References

- Jackson, Rebecca H and Fiammetta Straneo, 2016. Heat, salt, and freshwater budgets for a glacial fjord in Greenland, *Journal of Physical Oceanography*, **46**(9), 2735–2768.
- Kirkham, James D, Nick J Rosser, John Wainwright, Emma C Vann Jones, Stuart A Dunning, Victoria S Lane, David E Hawthorn, Mateusz C Strzelecki and Witold Szczuciński, 2017. Drift-dependent changes in iceberg size-frequency distributions, *Scientific Reports*, **7**(1), 1–10.
- Marko, John R, 1996. Small icebergs and iceberg fragments off Newfoundland: Relationships to deterioration mechanisms and the regional iceberg population, *Atmosphere-Ocean*, **34**(3), 549–579.
- Scheick, Jessica, Elyn M Enderlin and Gordon Hamilton, 2019. Semi-automated open water iceberg detection from Landsat applied to Disko Bay, West Greenland, *Journal of Glaciology*, **65**(251), 468–480.